# The Effectiveness of a Dyadic Pain Management Program for Community-Dwelling Older Adults with Chronic Pain: A Pilot Randomized Controlled Trial

**DOI:** 10.3390/ijerph17144966

**Published:** 2020-07-09

**Authors:** Ziyan Li, Mimi Tse, Angel Tang

**Affiliations:** School of Nursing, The Hong Kong Polytechnic University, Hung Hom, Hong Kong 999007, China; 18092892g@connect.polyu.hk (Z.L.); sk-angel.tang@polyu.edu.hk (A.T.)

**Keywords:** dyadic pain management, chronic pain, older adult, informal caregiver

## Abstract

Background: Chronic pain is a major health problem among older adults and their informal caregivers, which has negative effects on their physical and psychological status. The dyadic pain management program (DPMP) is provided to community-dwelling older adults and informal caregivers to help the dyads reduce pain symptoms, improve the quality of life, develop good exercise habits, as well as cope and break the vicious circle of pain. Methods: A pilot randomized controlled trial was designed and all the dyads were randomly divided into two groups: the DPMP group and control group. Dyads in the DPMP group participated in an 8-week DPMP (4-week face-to-face program and 4-week home-based program), whereas dyads in the control group received one page of simple pain-related information. Results: In total, 64 dyads participated in this study. For baseline comparisons, no significant differences were found between the two groups. After the interventions, the pain score was significantly reduced from 4.25 to 2.57 in the experimental group, respectively. In the repeated measures ANOVA, the differences in pain score (F = 107.787, *p* < 0.001, *d* = 0.777) was statistically significant for the group-by-time interaction. After the interventions, the experimental group participants demonstrated significantly higher pain self-efficacy compared with the control group (F = 80.535, *p* < 0.001, *d* = 0.722). Furthermore, the elderly increased exercise time significantly (F = 111.212, *p* < 0.001, *d* = 0.782) and reported developing good exercise habits. Conclusions: These results provide preliminary support for the effectiveness of a DPMP for relieving the symptoms of chronic pain among the elderly.

## 1. Introduction

Chronic geriatric pain is defined as an unpleasant sensory and emotional experience affecting persons over the age of 65 for more than 3 months, and which is associated with actual and potential tissue damage that is noncancerous in nature or is described in terms of such damage [1]. Chronic geriatric pain is a common public health concern. Chronic pain prevalence among community-dwelling older adults is approximately 40% in Hong Kong [2]. As the world’s population is aging rapidly, the number of older adults with chronic pain will likely continue to increase. Chronic pain can have a series of negative effects on individuals, both physical and psychological, causing them to lose the ability to care for themselves and to have a lower quality of life. If left untreated or undertreated, pain can grow as a result of adverse outcomes, including immobility, depression, anxiety, stress, social isolation, cognitive impairment, falls, and sleep and appetite disturbances [3,4,5,6], eventually increasing the economic burden on the sufferer’s family and on society [7]. The primary healthcare system will face severe challenges and enormous pressure in managing pain in the elderly. Therefore, it is necessary to explore efficient and cost-effective ways to manage chronic pain in community-dwelling older adults.

Some reviews examined a variety of pharmacological and non-pharmacological pain management programs for older adults, which have been carried out to explore their efficacy [8,9,10,11,12,13]. Since chronic geriatric pain lasts for a long time and interacts with other age-related diseases [14], the main objective is to relieve the pain and pain-related symptoms, rather than to completely eliminate the pain. Specifically, healthcare providers conduct pharmaceutical or non-pharmaceutical treatments for older adults with chronic pain to boost their psychology, improve their functional well-being, and enhance their quality of life. Even though analgesics remain the primary approach to managing pain, some older adults worry about the possibility of adverse drug reactions [15]. Besides this, homeostenosis in aging has negative effects on the absorption, excretion, and response to drugs, which can cause the pain relief for the elderly from these drugs to be less than expected [16,17]. Due to these factors, more and more non-pharmacological pain management strategies are being used to deal with chronic pain [18]. In a literature review [13], researchers investigated the evidence on the efficacy of physical exercise and educational interventions for pain relief among community-dwelling older adults. A total of seven studies reported that older adults with chronic pain who participated in physical exercise programs can significantly reduce their pain, enhance their physical function, as well as improve their well-being and self-efficacy. The greater pain self-efficacy means that the elderly has confidence in the ability to deal with the symptoms, stresses, or limitations associated with a pain condition. In addition, the Centers for Disease Control (CDC) guidelines recommend physical exercise therapy as a non-pharmacologic strategy to address chronic pain [19]. There were five studies on pain educational interventions, including one online program and four face-to-face programs. Pain education cause older adults to more clearly understand their responses to pain, which led to significantly less pain. Education programs typically include the symptoms of pain, type of pain, physical and psychological effects of pain, assessment instruments, drug and non-drug therapies, and coping strategies [20,21].

Pain management for older adults differs significantly from that for younger adults and is more challenging [22]. Because of limited mobility and poor memory, the ability of aging people to carry out the activities of daily life are hindered, and older adults are less likely to take part in exercise and social events [23]. Compared with the general population, poorer attendance rates and poorer compliance with the interventions pose obstacles to the management of chronic geriatric pain [24]. In addition, some older adults have had little education or have some degree of cognitive impairment, which can result in poor communication between healthcare providers and patients and raise barriers to the assessment and management of pain [22]. It also can be difficult for elderly people to understand pain-related theoretical knowledge and accept online or digital pain interventions. In previous studies, elderly people who took part in pain management programs often failed to develop good exercise habits and fell back into old habits of inactivity after finishing the program [25,26]. Therefore, it is important to develop interventions that can be sustained over time.

To address the above problems, consideration was given to implementing a dyadic pain management program (DPMP). In Hong Kong, more than 90% of older adults were living in domestic households; of those, 25.2% were living only with their spouse, 29.0% were living with a spouse and children, and 19.5% were living only with their children [27]. That means that older adults are mainly being cared for by informal caregivers. An informal caregiver is a family member or close friend who has taken responsibility for the physical and emotional needs of a person who cannot entirely care for himself or herself because of advanced age, illness, dementia, or disability without an income [28]. Informal caregivers of older adults with chronic pain have a wide range of responsibilities that normally include helping the elderly in activities of daily living, reminding them to take their medications and to do exercises to relieve pain symptoms, taking them to the doctor when necessary, communicating with them, providing emotional support, and encouraging them to engage in social activities [29]. When informal caregivers take part in physical exercise programs or educational programs with the elderly, they become an important bridge between older adults and healthcare providers. Caregiver education is especially important in caring for the elderly [30]. Interventions targeting informal caregivers can improve their knowledge and coping skills, with the result that they are able to take better care of the elderly and experience their caring role more positively [31]. Each older adult and informal caregiver make up a one-to-one group, in which the informal caregiver is considered the supervisor, guardian, partner, assessor, commentator, and prompter of the older adult. Providing caregivers with education can be a good opportunity for informal caregivers to communicate with others who may have similar caregiving burdens and problems. Feedback from informal caregivers has shown that a dyadic intervention can be a good opportunity for the caregivers to explore activities in which they share an interest with their patient [31]. The Theory of Dyadic Illness Management moves beyond a discussion of the ways in which the individual patient and care partner respond to illness and focuses extensively on the dyad as an interdependent team [31]. In a trial based on that framework, older adults and their informal caregivers exercised in tandem and were required to interact physically and verbally as a team during the exercises, with the ultimate goal of improving the physical and mental health of both members of the dyad [32,33].

Dyadic interventions for clients with depression or dementia have been examined [34,35]. However, to the best of our knowledge, only two studies have used patient/caregiver dyads in a pain management intervention [36,37]. A study by Keefe et al. targeted adults with osteoarthritic knee pain and involved 72 dyads (patients and their spouses). A study by Abbasi et al. targeted adult patients with chronic low back pain and involved 36 dyads (patients and their spouses). Both intervention studies consisted of spouse-assisted pain coping skills and exercise training. Both studies demonstrated improvements in pain intensity, psychological distress, and marital adjustment, as well as a decrease in pain catastrophizing. Existing dyadic pain management programs only focus on the spousal relationship, and all of the participants have been adults in general, rather than elderly people in particular. The objectives of this study were to develop a DPMP for older adults with chronic pain and their informal caregivers, to (1) evaluate the effectiveness of a DPMP in reducing pain and psychological health symptoms, improving pain self-efficacy, and quality of life in older adults; and (2) explore the acceptability and satisfaction of informal caregivers and older adults in participating in the DPMP.

## 2. Materials and Methods

### 2.1. Study Design

A pilot randomized controlled trial was conducted. Computer-generated random numbers were used to divide the dyads into either the experimental group or the control group. Each dyad was given a sealed envelope with the number of a research assistant who had no knowledge of this subject, and who was given this random number generated by a computer. Using the sequentially numbered, opaque, sealed envelopes, allocation concealment was carried out by a research assistant who had no knowledge of this subject. To reduce the dropout rate, the dyads were blinded to the grouping situation. The experimental group received an 8-week DPMP, which included a 4-week face-to-face program and a 4-week home-based program delivered via digital tools. The control group received one page of simple pain-related information and would be invited to join in 4-week face-to-face program and given an exercise book, in which the content is the same as the experimental group after the end of the 8-week intervention. The flow of this study is shown in Figure 1.

Ethical approval was obtained from the institutional review board of the Hong Kong Polytechnic University before the start of this study (reference number: HSEARS20190617003, 28 June 2019). The trial was registered with the Clinical Trials Centre of the Hong Kong Polytechnic University (NCT04106271, 28 June 2019). Written informed consent was obtained from all of the participants in the study.

### 2.2. Participants

This research was a dyadic pain management intervention. The main target population was older adults with chronic pain. A pair of older adults and their main informal caregivers was regarded as a dyad. The dyads were recruited from four community elderly centers in Hong Kong between July 2019 and November 2019. These community activity centers offer free services to community-dwelling residents and regularly hold health lectures. Social workers put up posters introducing the program on community bulletin boards, to recruit older adults and informal caregivers who might be interested in taking part. Older adults were screened according to the following inclusion criteria: (1) ≥60 years of age; (2) non-cancer pain duration ≥3 months; (3) pain score ≥2 assessed on a 0 to 10 numeric rating scale (NRS); (4) mainly cared for by informal caregivers; (5) able to understand Cantonese; (6) have sufficient behavioral abilities to take part in a light exercise and stretching program; (7) has an informal caregiver who owns a mobile phone and who can access the internet; and (8) able to join in whole program with their informal caregivers. Older adults were excluded if they had undergone medical or surgical treatment in the past two months, or had a history of serious organic disease, a malignant tumor, loss of consciousness, a mental disorder, a drug addiction problem, or were on scheduled pain medications.

This was a pilot study; thus, it was decided that 60 participants, 30 per group, would be recruited. When a significant difference is unknown and when investigators want to calculate the sample size for a larger study, 30 to 40 patients per group is necessary [38].

A total of 82 eligible dyads were recruited. Each potential dyad received an information sheet with details of the research. Eighteen dyads refused to take part, either due to a lack of time or other personal reasons. Eight dyads withdrew after randomization because of personal reasons, such as a tight schedule, a previous engagement, having been out of contact during the data collection period, and so forth, leaving 32 dyads in the DPMP group and 32 dyads in the control group (total *n* = 64) for the analysis.

### 2.3. Intervention

A DPMP is an 8-week, single-group program. The DPMP included 4 weeks (Weeks 1 to 4) of center-based, face-to-face activities and 4 weeks (Weeks 5 to 8) of home-based and digital-based activities delivered via a WhatsApp group. Timely make-up sessions were arranged for those who were unable to attend the scheduled session. More details are shown in Figure 2.

For the face-to-face part, the DPMP began with 20 min of physical exercises supervised by a research assistant, followed by 20 min of pain management education, including pain-related theoretical knowledge, caring-related coping skills, the negative effects of chronic pain, the use of pharmacological and non-pharmacological intervention strategies for pain management, and demonstrations of non-drug pain management methods. Communication skills were taught, and the participants were encouraged to practice various pain-relief methods at home.

For the home-based part, an exercise book was given to each dyad to guide them in performing daily exercises at home at Week 4, the end of the face-to-face program. The researcher showed how to use this exercise book at home. The exercise book contains detailed images of each step in the exercises, which are the same exercises as those performed by the exercise dyads in the community center. This book was given to each dyad to ensure that the participants would know how to perform the exercises at home. The exercise book includes the whole exercise process: warming up, breathing exercises, stretching exercises, strength exercises, balance exercises, and relaxation exercises. Details of each step of the exercises are presented in Figure 3. The researcher sent the participants reminders via WhatsApp and recorded the completion rate.

The use of a WhatsApp group (digital-based activities): Each dyad joined a WhatsApp group to receive teaching materials and videos of the physical exercises learned in class, to practice at home. Each dyad was encouraged and reminded to practice the 30-min exercises together at home and to make entries in the WhatsApp group, as well as to record which of the various types of non-pharmacological methods they used to relieve their pain and their perception of the effectiveness of those methods.

The content was vetted by five experts in pain management research, including three university professors whose area of research is pain management, and two registered nurses working in hospital pain clinics who have a tremendous amount of experience in pain management. The experts provided feedback on how to enrich the details of this study and expand the questions and strategies. After multiple rounds of revisions, the experts validated the contents of the manual. The content validity index score was 0.95. Participants in the control group were given the usual care and a pain management pamphlet

### 2.4. Data Collection

Outcomes were measured at two time points throughout the study: T0 at baseline before the intervention, and T1 at week eight when the DPMP group completed the entire intervention. The dyads answered all of the demographic and caregiving-related questions in T0. The pain-related situations (pain severity, pain interference, and pain self-efficacy), psychological parameters, and quality of life of the older adults were measured at T0 and T1. Tests on the acceptability of the program to the dyads and their satisfaction with it were evaluated immediately at the end of intervention. The data were collected by a research assistant who was blinded to the dyads’ group assignments.

### 2.5. Outcome Measures

#### 2.5.1. Brief Pain Inventory—Chinese Version (BPI-C)

Pain severity and pain interference with the activities of life in the previous 24 h were assessed by BPI-C [39]. The total scores for pain severity (4 items) and interference (7 items) were measured with a 11-point scale (from 0 = no pain/interference to 10 = worst pain/interference imaginable), where higher scores represent more violent pain. The Cronbach’s α for the pain severity and pain interference items were 0.83–0.89 and 0.90–0.91, respectively, and thus have a good and acceptable test–retest reliability [40].

#### 2.5.2. Pain Self-Efficacy Questionnaire—Chinese Version

Self-efficacy for pain was assessed by using the 10-item Pain Self-Efficacy Questionnaire (PSEQ) with a 7-point scale (from 0 = not at all confident to 6 = completely confident) [41]. Pain self-efficacy is generally defined as person’s confidence in carrying out daily activities, despite suffering from pain. The Chinese version of the PSEQ was used in our study, with higher scores indicating a higher level of self-efficacy for pain. The Cronbach’s α was 0.95 and the test–rest reliability coefficient was 0.75 [42].

#### 2.5.3. The World Health Organization Quality of Life—BRIF

Quality of life was measured by using the 26-item World Health Organization Quality of Life—BRIF (WHOQOL-BRIF) with a 5-point Likert scale (from 1 = not at all to 5 = completely), including four domains (physical health, psychological health, social relationships, and environment) [43]. A previous study has shown that WHOQOL-BRIF has good reliability and validity [44].

#### 2.5.4. Depression Anxiety Stress Scales-21—Chinese Version

Depression, anxiety, and stress status were assessed to evaluate each dyad’s mental health, using the Depression Anxiety Stress Scales 21-item (DASS-21), a self-administered psychological tool. Each part has seven items on a 4-point Likert scale (from 0 = did not apply to me at all to 3 = applied to me very much, or most of the time) [45]. This assessed the common depression, anxiety, and stress symptoms over the past week. The Cronbach’s α was 0.912 and the test–retest Pearson correlation coefficient was 0.751 [46].

#### 2.5.5. Satisfaction and Acceptability

The acceptability of the dyadic intervention to the participants and their satisfaction with it was measured when the DPMP group completed the entire intervention. Some open-ended questions were asked, including, “Did you think the pain-related knowledge and coping skills taught in the program is useful and sufficient ?”; “Which parts of content could have been changed or improved?”; “Participating in this intervention, what benefits did this DPMP bring to your life?”; “Did you feel the dyadic home-based exercise interesting and could you stick to it?”; “Was it convenient and easy to take part in this intervention via digital tools”; and “Did the interactive activities and exercises relieve your pain symptoms and psychological problems?”. These questions had been used in previous studies to assess the acceptability of an intervention program and the participants’ satisfaction with it [47,48].

### 2.6. Statistical Analysis

SPSS statistics (IBM Corp, Armonk, NY, USA) version 22 was used for the statistical analyses. Descriptive statistics were used to describe the demographic characteristics, pain-related variables, and caring-related situations. The mean (M), standard deviation (SD), and frequency were calculated. Data were examined with the intention to treat (ITT) analysis (all randomized participants, *n* = 64). The SPSS Missing Value Analysis was used to impute missing post-intervention (*n* = 8) data with the expectation-maximization method [49]. The Shapiro–Wilk test was used to assess whether the data were normally distributed. To examine the differences in demographic characteristics and outcome variables between the two groups, at baseline, a Chi-square test and independent sample t-test were applied. The repeated measure ANOVA was used to explore how outcomes had changed over time, between groups, and the interaction between time and group. Repeated measures models with two intervention variables (DPMP group, control group) and two time points (T0, T1) were used to test the efficacy of this dyadic intervention in reducing pain intensity, pain interference, depression, anxiety, stress, improving pain self-efficacy, quality of life, and increasing average exercise time (per week). Spearman correlation analyses were adopted to test whether there was any association between the frequency with which the digital tools of the intervention were used and the measured factors. Open-ended treatment satisfaction and acceptability questions were analyzed using a conventional content analysis. Open-ended questions were recorded by one research assistant and the transcripts were cross-checked for accuracy by another two research assistants. The finalized transcripts were independently coded by two researchers, and important manifest contents and latent meanings in the data were identified. Major themes were identified based on the final codebook and discussed with all team members to reach final approval. Finally, a set of categories and subcategories with supporting verbatim data were generated to describe the acceptability and satisfaction of the DPMP. Statistical significance was set by a two-tailed test and a *p*-value of <0.05 was considered statistically significant.

## 3. Results

### 3.1. Baseline Characteristics

The demographic and clinical characteristics of the two groups of older adults are presented in Table 1. The DPMP group and the control group had similar demographic and clinical profiles at baseline. The mean age of the older adults was 71.7 years (SD = 14.6) and the majority of the older adults had attained a middle school level of education (47%), were female (73%), married (75%), and were being cared for by a spouse (58%) and children (36). The prevalence of hypertension and diabetes was highest, which was 47% and 39%, respectively. The two groups did not differ significantly in their demographic and clinical characteristics.

### 3.2. Pain Severity and Pain Interference

As seen in Table 2 and Table 3, the outcome variables were normally distributed, as evaluated by a Shapiro–Wilk test. Significant differences in pain score (F = 107.787, *p* < 0.001, *d* = 0.777) were found for the group-by-time interaction. For comparisons before intervention, there were no significant differences between the DPMP group and control group. After the intervention, the experimental group had a significantly lower pain score compared with the control group, for which the mean difference was −1.438 (95% CI: −1.974−0.901, F = 29.871, *p* < 0.001). The pain severity of the DPMP group was also significantly different between pre-treatment and post-treatment (F = 265.809, *p* < 0.001, *d* = 0.896), and a significant difference was also observed in the control group (F = 41.015, *p* < 0.001, *d* = 0.570). The experimental group showed a better intervention effect.

A statistically significant difference in pain interference was not found for the group-by-time interaction. For pre-intervention and post-intervention comparisons, the differences between the two groups was not statistically significant. Pain interference in the DPMP group was significantly reduced (F = 99.863, *p* < 0.001, *d* = 0.763).

### 3.3. Depression, Anxiety, and Stress

The changes in the depression, anxiety, and stress levels of the older adults are shown in Table 2 and Table 3. There were no significant changes in the depression subscales and anxiety subscales at the between-group and within-time comparisons. The stress subscales had a significant decrease within-time (F = 32.700, *p* < 0.001, *d* = 0.513).

### 3.4. Pain Self-Efficacy

The pain self-efficacy results of the older adults are reported in Table 2 and Table 3. The between-group difference was not statistically significant at baseline. For post-intervention comparisons, the elderly in the DPMP group had a significantly higher pain self-efficacy than those in the control group (*p* < 0.001). A significant improvement in pain self-efficacy (36.53 to 47.25) was observed in the experimental group after the program (F = 111.034, *p* < 0.001, *d* = 0.944). There were no statistically significant changes in the control group. The between-group differences were not significant over time.

### 3.5. Quality of Life (QoL) and Exercise Time

For post-intervention comparison, the elderly in the DPMP group illustrated a significantly higher level of QoL in the physical health subscales, psychological health subscales, and social relationship subscales (*p* < 0.001) than the control group. Table 2 and Table 3 show that the physical health subscales, psychological health subscales, and social relationship subscales were significantly improved over time. The between-group difference was also statistically significant. Moreover, after the repeated measures ANOVA, statistically significant differences in the physical health subscales (F = 92.711, *p* < 0.001), psychological health subscales (F = 14.783, *p* = 0.001), and social relationship subscales (F = 87.904, *p* < 0.001) were found for the group-by-time interaction. However, no significant change in the environment subscales was found in both groups.

Compared with the control group, older adults in the DPMP group had a significant improvement in average exercise time (*p* < 0.001). The group-by-time interaction was statistically significant (F = 111.212, *p* < 0.001, *d* = 0.782). The between-group differences were non-significant.

### 3.6. Use of Digital Tools

The total number of WhatsApp messages was 3429, including 2378 sent by the researchers and 1051 sent by the dyads. The average number of messages sent during the whole program was 107 per dyad. Details are shown in Table 4. More than 70% of the messages were read within 1 h. The correlation between the frequency with which the digital tool was used and the outcomes is demonstrated in Table 5. A significant correlation was observed between the frequency with which the digital tool was used and pain intensity, QoL–physical health, and average exercise time.

### 3.7. Learning Performance, Satisfaction, and Acceptability

All in all, most of the participants reported that they were satisfied with this DPM program, and felt that it was worth spending time on. Moreover, the participants showed a willingness to recommend this program to others. The answers to the open-ended questions also showed that the program was acceptable to the participants and that they were satisfied with it: “I can relieve my stress and I enjoy doing the exercises”; “the knowledge is useful”; and “will recommend it to others”. More detailed information is shown in Table 6.

## 4. Discussion

The objectives of this study were to evaluate the efficacy of a DPMP, its acceptability to the participants, and their satisfaction with it. Upon completing the DPMP, the older adults experienced a significant reduction in the intensity of their pain. In addition, pain interference, pain self-efficacy, QoL (physical health, psychological health, and social relationships), and average exercise time improved significantly in the experimental group. A significant positive correlation was demonstrated between QoL (physical health) and average exercise time, as well as with the frequency with which the digital tool was used. There was also a significant negative correlation between pain intensity and the frequency with which the digital tool was used, with high doses being more effective. The results demonstrated that this DPMP was acceptable, obtaining high satisfaction levels and with the dyads willing to recommend this pain management to others.

The major findings suggest that this DPMP has the potential to relieve the pain symptoms of older adults. The results of a significant decrease in pain intensity and pain interference are in keeping with previous studies [50,51,52,53,54,55]. However, the improvement of psychological parameters was not obvious. The intervention is composed primarily of physical exercise and supported by health education. The teaching materials used in the program focus on the theoretical level, which are difficult to understand, even for some informal caregivers. A previous study showed using picture books as intervention tools was helpful to improve the attention and executive function in community-dwelling older adults [56]. Another reason might be that the length of the intervention was too limited to improve the psychological parameters significantly [55]. Informal caregivers could easily carry out the home-based exercise, but it is hard for them to deal with the psychological and emotional problems without professional knowledge.

Most informal caregivers provided feedback that they were also suffering from chronic pain and got benefit from the pain management. Caregiver burden is what caregivers have perceived whilst caregiving, which has had an adverse effect on their emotional, social, financial, physical, and spiritual functioning [52]. Some studies show that, compared with non-caregivers, caregivers often suffered from psychological, behavioral, and physiological effects that can contribute to impaired physical and psychological status because of caring [57,58]. They have to pay extra time and experience and even modify their lifestyle to meet their recipients’ demands, including limiting leisure activity, reducing the time on outdoor activities, and leaving little time on performing health assessments [59,60]. Many family caregivers had to take leave from work to care for older adults and even resigned, but in this way the economic burden of the family would increase sharply [61,62]. Sometimes when they provided care for their older patients having no improvement, they felt helpless, sad, and frustrated [63]. More than 7% of female informal caregivers had thoughts of suicide associated with a lack of social support, social integration, and huge caregiver burden, which is much higher than the normal [64]. Dyadic pain education can relieve pain symptoms in both older adults and their informal caregivers [65,66]. Educational initiatives to improve the pain management knowledge of dyads ultimately improved their practices and their quality of life [67]. As the population continues to age, the elderly always undertake caregiving roles for sick spouses or other relatives [68]. Old adults and their informal caregivers faced many of the same or similar health problems. As such, it is important to develop better DPMPs.

Note that a significant correlation was observed between the frequency with which the digital tool was used and the physical parameters. Previous studies reported that the more frequently they used digital tools, the better the outcome achieved—being milder pain, better psychological well-being, and higher self-efficacy [21,69]. The high usage of digital tools is presumably due to reminders we sent via WhatsApp. The messages were transmitted to informal caregivers regularly, to encourage and support them to take out the older adults to do physical exercise, as guided by exercise book, and keep reviewing the pain-related theoretical knowledge and coping skills taught in the face-to-face program. The researchers also kept in touch with the informal caregivers and could constantly answer questions and solve the problems. Besides, short videos and audio clips were sent to informal caregivers to help clarify and demonstrate a complete set of exercise. A short video combined with an exercise book is a more convenient and efficient way to guide older adults to do exercise and develop good exercise habits. Meanwhile, it is a good way to use a fragment of time. In previous studies, researchers used emails, phone follow-up, and reminder messages to prompt the participants in the self-management programs to learn the teaching materials and carry out the interventions, and in this way could effectively reduce the withdrawal rate and facilitate the completion rate [70,71].

In our study, we used digital tools, which are innovative and cost-effective pain management strategies. These features are important because of the aging population and the high prevalence of pain [72,73]. There are two common digital health techniques: electronic health (eHealth), including the professional, low-cost, and efficient telemedicine platforms, as well as computer-based big medical datasets and communication systems for real-time updates of health-related information; or mobile health (mHealth), which is defined as a health-related practice supported by mobile devices. Despite the acceptability of digital tools among the elderly not being good, this general tendency is being reversed, and the health management of older people using digital health techniques will increase with the ubiquity of this technology, especially when making digital products for the elderly [62]. According to the current situation, it is hard for older adults to join in the digital-based pain management intervention independently, because of limited understanding of how the device functioned, and the occasional technical malfunction [64]. Therefore, in this study, older adults and their informal caregivers were regarded as a dyad and joined the program together. It is convenient for older adults to use digital tools with the help of informal caregivers. At the same time, when the informal caregivers received the reminders and teaching materials, they could share these with the older adults. The caregivers became a bridge between the older adults and the researchers, allowing them to collaborate.

This research has the following strengths. Firstly, it was the first pilot randomized controlled trial to evaluate a DPMP for older adults with chronic pain and their informal caregivers. Secondly, the completion rate of the intervention and questionnaires performed well. Thirdly, a digital-based interactive session was included in the DPMP, and the dyads’ learning performance was assessed. Besides, the correlation between the frequency with which digital tools were used and the outcomes was tested. Finally, the satisfaction and acceptability were explored by a qualitative analysis of the dyads’ feedback.

Because of limited research time, we just collected one time point’s (T1) data during the whole intervention. However, this was too infrequent—measurements needed to be taken more often to capture the effects of the intervention. In addition, this study lacked a follow-up to determine whether the benefits that were observed can be sustained over a longer period. Secondly, this study had a small sample size, which we plan to enlarge in a future study. This study used self-reported questionnaires; as a result, because of difficulties in understanding and writing for older adults, some questionnaires were finished with the help of their informal caregivers. Thus, personal bias cannot be completely ruled out. Additionally, the physical improvements, demographic, and clinical characteristics of the informal caregivers were also not measured in this study.

## 5. Conclusions

Our findings highlight the significant potential of dyadic pain management programs to enhance healthy living as well as to reduce pain. Further promotion of such programs to the public can help more people. Based on our results, this study can be used as a reference for subsequent evidence-based clinical nursing practices and home-based interventions. With the wide application of electronic networks and development of mobile health, it is hoped that an online, home-based, remote monitoring approach can be expanded to treat other pain-related problems in other regions and countries in future investigations.

## Figures and Tables

**Figure 1 ijerph-17-04966-f001:**
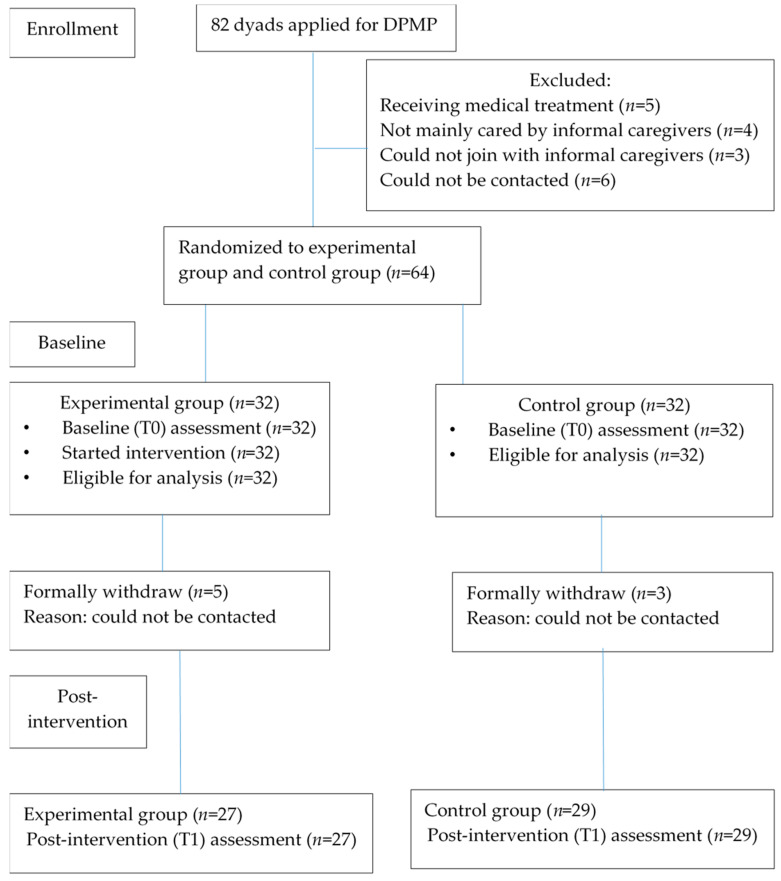
The flow of this study.

**Figure 2 ijerph-17-04966-f002:**
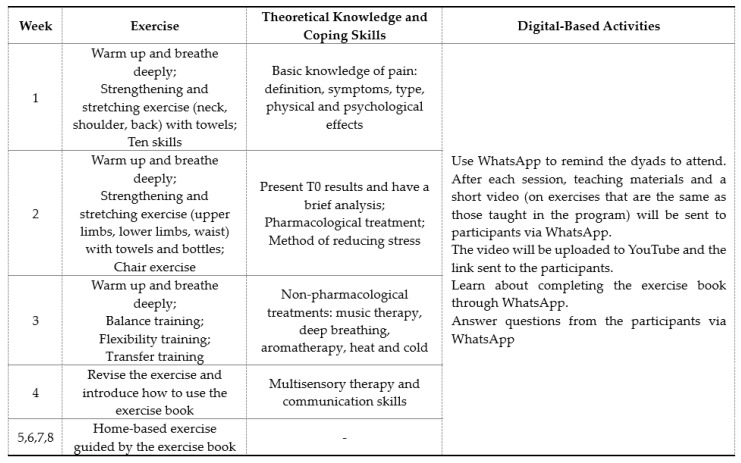
The contents of the dyadic pain management program (DPMP).

**Figure 3 ijerph-17-04966-f003:**
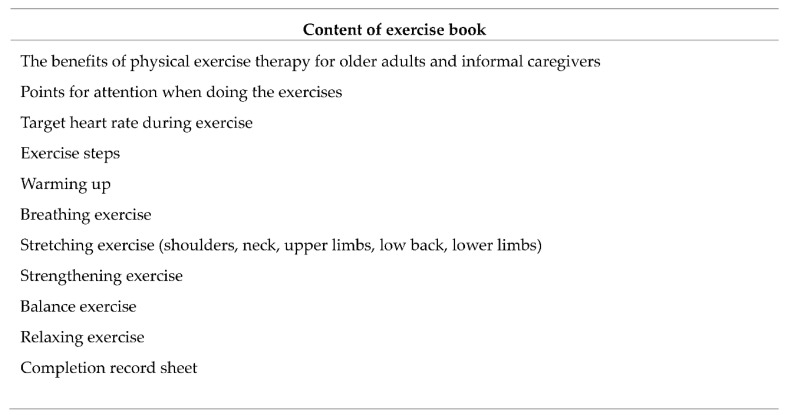
The contents of the exercise book.

**Table 1 ijerph-17-04966-t001:** Baseline demographic and clinical characteristics of the older adults.

Variables	Total (*n* = 64)	Experimental Group (*n* = 32)	Control Group (*n* = 32)	χ^2^	*p*
Gender, *n* (%)				0.34	0.67
Female	47 (73)	23 (72)	24 (75)		
Male	17 (27)	9 (28)	8 (25)		
Age ^a^				0.98	0.55
Mean (SD)	71.7 (14.6)	70.7 (15.8)	72.6 (13.7)		
Range	61–92	61–92	64–89		
Marital status, *n* (%)				1.65	0.23
Single	0 (0)	0 (0)	0 (0)		
Married	48 (75)	23 (72)	25 (78)		
Divorced	8 (13)	5 (16)	3 (9)		
Widowed	8 (13)	4 (13)	4 (13)		
Education, *n* (%)				0.87	0.45
Less than high school	12 (19)	7 (22)	5 (16)		
High school	30 (47)	13 (41)	17 (53)		
College certificate	17 (27)	9 (28)	8 (25)		
University or above	5 (8)	3 (9)	2 (6)		
Monthly income, *n* (%)				1.21	0.65
<HK $10,000	47 (73)	23 (72)	24 (75)		
HK $10,000–20,000	14 (22)	8 (25)	6 (19)		
>HK $20,000	3 (5)	1 (3)	2 (6)		
Relationship with caregivers, *n* (%)				1.09	0.38
Spouse	37 (58)	18 (56)	19 (59)		
Parent	23 (36)	12 (38)	11 (34)		
Child	3 (5)	1 (3)	2 (6)		
Other	1 (2)	1 (3)	0 (0)		
Chronic diseases ^b^, *n* (%)					
Heart disease	16 (25)	9 (28)	7 (22)	0.29	0.59
Diabetes	25 (39)	13 (41)	12 (38)	0.54	0.89
Hypertension	30 (47)	16 (50)	14 (44)	0.43	0.78
Tracheal disease	12 (19)	5 (17)	7 (22)	0.58	0.48
Cataract	9 (14)	6 (19)	3 (9)	1.32	0.13
Stroke	7 (11)	3 (9)	4 (13)	0.32	0.67
Arthritis	17 (27)	8 (25)	9 (28)	0.37	0.57
Gout	8 (13)	3 (9)	5 (17)	0.87	0.43
Other chronic disease	7 (11)	3 (9)	4 (13)	0.46	0.78

^a^ The statistics were calculated by independent-samples *t*-tests. ^b^ The participants could choose more than one choice. Data with a *p* value < 0.05 indicate statistical significance.

**Table 2 ijerph-17-04966-t002:** Pre-post outcome measurements.

Variables	Group	T0 Mean ± SD	T1 Mean ± SD	*F*	*p* ^a^	*d*
Pain score				107.787	<0.001	0.777
	Experimental group	4.25 ± 1.05	2.57 ± 0.97			
	Control group	4.43 ± 1.09	4.01 ± 1.08			
Pain interference				2.153	0.152	0.056
	Experimental group	2.85 ± 1.52	1.88 ± 1.29			
	Control group	2.89 ± 1.70	2.22 ± 1.38			
Pain self-efficacy				80.535	<0.001	0.722
	Experimental group	36.53 ± 12.62	47.25 ± 8.46			
	Control group	36.59 ± 11.58	37.72 ± 11.97			
DAS—Depression				0.704	0.408	0.022
	Experimental group	8.75 ±6.79	8.00 ± 6.66			
	Control group	8.63 ± 6.47	8.31 ± 6.66			
DAS—Anxiety				0.892	0.313	0.016
	Experimental group	10.26 ±6.32	9.84 ± 5.73			
	Control group	10.71 ±6.94	10.48 ± 6.12			
DAS—Stress				15.360	<0.001	0.331
	Experimental group	11.50 ±7.55	8.43 ± 6.60			
	Control group	11.00 ±6.18	10.75 ± 5.90			
QoL—Physical health				92.711	<0.001	0.749
	Experimental group	47.94 ± 9.36	67.44 ± 11.08			
	Control group	48.94 ± 9.19	51.25 ± 8.18			
QoL—Psychological health				14.783	0.001	0.323
	Experimental group	54.19 ± 10.87	66.47 ± 11.27			
	Control group	53.44 ± 7.79	53.56 ± 7.30			
QoL—Social relationships				87.904	<0.001	0.712
	Experimental group	52.43 ± 9.43	67.94 ± 12.48			
	Control group	54.95 ± 10.38	57.24 ± 11.22			
QoL—Environment				0.805	0.364	0.017
	Experimental group	57.65 ± 7.27	58.24 ± 8.43			
	Control group	56.85 ± 8.16	57.57 ± 9.04			
Average exercise time (min/per week)				111.212	<0.001	0.782
	Experimental group	81.03 ± 37.84	134.38 ± 31.79			
	Control group	83.53 ± 38.47	87.97 ± 37.39			

^a^ The statistics were calculated by two-way mixed ANOVA (interactive effect). Data with a *p*-value < 0.05 indicate statistical significance.

**Table 3 ijerph-17-04966-t003:** Results of the repeated measures ANOVA for the outcome variables.

Variables ^a^	Within-Groups			Between-Groups		
	*F*	*p*	*d*	*F*	*p*	*d*
Pain score	290.005	<0.001	0.903	10.244	0.003	0.248
Pain interference	102.788	<0.001	0.768	0.415	0.524	0.013
Pain self-efficacy	105.882	<0.001	0.774	3.089	0.089	0.091
DAS—Depression	3.334	0.077	0.097	0.004	0.948	0
DAS—Anxiety	2.198	0.087	0.104	0.012	0.215	0.047
DAS—Stress	32.7	<0.001	0.513	0.343	0.562	0.011
QoL—Physical health	158.976	<0.001	0.837	10.469	0.003	0.252
QoL—Psychological health	11.713	0.002	0.274	14.474	0.001	0.318
QoL—Social relationships	134.854	<0.001	0.812	10.784	0.002	0.267
QoL—Environment	2.765	0.081	0.098	0.007	0.854	0.002
Average exercise time (min/per week)	179.652	<0.001	0.853	6.293	0.018	0.169

^a^ The statistics were calculated by repeated measures ANOVA. Data with a *p*-value < 0.05 indicate statistical significance.

**Table 4 ijerph-17-04966-t004:** Use of digital tools.

Variables	Total	Average (Per Participant)
Number of WhatsApp messages (including video and voice)		
Total number of WhatsApp messages	3429	127.1
Sent by researchers	2378	88.1
Sent by participants	1051	38.9
Messages read rate		
Message read within 30 min	785	29
Message read within 1 h	982	36.4
Message read within 2 h	304	11.3
Message read within 3 h	237	8.8
Message read within 1 day	42	1.6
Message read after more than 1 day	28	1

**Table 5 ijerph-17-04966-t005:** Correlation between the frequency of using digital tools and the outcome variables.

Variable	Post-Intervention (T1)	
	r ^a^	*p*
Pain score	−0.572	0.031
Pain interference	−0.135	0.124
Pain self-efficacy	0.083	0.24
Depression	−0.076	0.34
Anxiety	−0.053	0.29
Stress	−0.092	0.48
Physical health	0.341	0.045
Psychological health	0.145	0.067
Social relationships	0.089	0.15
Environment	0.067	0.62
Average exercise time	0.689	0.004

^a^ The statistics were calculated using Pearson correlations. Data with a *p*-value < 0.05 indicate statistical significance.

**Table 6 ijerph-17-04966-t006:** Perspectives and experiences of informal caregivers and older adults.

Categories/Themes	Comments and Feedback from Dyads
Perceived benefits: helping both older adults and their informal caregivers	My pain is gone after joining the program with my mom This program effectively relieved the pain of the participants I feel happier and less lonely I can relieve my stress and enjoy doing the exercises
Communication	I can communicate with peers Caregivers can share their care-related problems and experiences with other caregivers I have more time to communicate with the elderly
Boosted my sense of self-worth	My mom recognized all the hard work I did to care for her and I was proud of myself The elderly and caregivers understand each other
Feedback on the content of the DPM program	I like the DPM program I’d like to take part in more programs like this one To improve the DPM program, e.g., extend the exercise time and add more interactive games, alternate a face-to-face program with a digital education program, reduce the complex introduction of professional knowledge and use more pictures WhatsApp messages can remind me do the exercises The short video is useful and I can watch it before I do the exercises

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
