# Peer review of "The Effectiveness of a Dyadic Pain Management Program for Community-Dwelling Older Adults with Chronic Pain: A Pilot Randomized Controlled Trial"

_ijerph, 2020, doi:10.3390/ijerph17144966_

Round 1

Reviewer 1 Report

The present study presents a dyadic exercise-based treatment program for older adults with pain. Even though the research topic is interesting, there are important flaws in the manuscript that should be addressed before it is suitable for publication. My comments can be found below.

Review

English should be revised throughout the text.

Abstract

The statement “The outcomes were measured” seems incomplete.

The word “also” is incorrect in “The pain intensity of the experimental group 24 was also significantly reduced among older adults and informal caregivers.”

Statistics on effect sizes would be needed to have an overview of the size of the differences found. The analyses made should be mentioned.

Introduction

The introduction is, in general, poorly organized and the paragraphs are not well connected. For example, the first paragraph appears to simply present a number of partly related statements. The same applies to the second paragraph (e.g., when statement “The most common sites of pain for older adults are the back, arms, hips, and 45 legs [13].” is presented) and so on.

The statement “have influence on physical and psychological aspects” is too vague.

The authors present previous research on dyadic treatment in pain settings. One of the contributions of the present work is the focus on older adults. Can the authors describe the age characteristics of participants in previous research to see whether differences with the present study are that significant? Also, a rationale for differentiating treatments for adults and older adults is missing. Why are previous treatments not applicable because of age differences? What is unique to older adults that required a different treatment and makes the results of past research not applicable? The authors should also mention additional treatment studies (“A Couple-Based Psychological Treatment for Chronic Pain and Relationship Distress” and “Adjunctive cognitive behavioural treatment for chronic pain couples improves marital satisfaction but not pain management outcomes”, to name some examples). The justification for the study, which is mostly related to the previous lines, is poorly described.

Materials and methods

The treatment is very focused on physical exercise. A rationale for doing this should have been emphasized in the introduction. Is the treatment evidence-based? I am thinking for example in the non-pharmacological treatment part. There is no evidence to CBT or ACT principles, for examples. I wonder which guidelines were followed to select the content to be included, as well as the outcomes selected (see IMMPACT or VAPAIN guidelines for outcomes in pain research).

Did the authors ask the control group about the satisfaction with treatment? Did they offer the control group to participate in the experimental condition at the end of the study for ethical reasons?

In the clinicaltrials registration the authors mention that measures would be taken at week 4, but this is not mentioned in the manuscript.

The active comparator is very poor in terms of the content provided, which should be mentioned as a study limitation (it is almost as a waiting list).

How were between-group differences in study variables evaluated? This is not mentioned anywhere in the data analysis plan.

Why are caregivers measured on pain severity, interference, and pain self-efficacy? Did they have chronic pain too?

Results

Chi-square and t values are required in Tables 1 and 2.

The statistics provided in the text from the results section do not follow recommended guidelines.

In Tables 3 and 6, statistics for between comparisons are needed (what is a repeated-measures MANOVA?).

In Table 5, the correlation should be made with change scores (from baseline to post-treatment).

The analyses made should be based on study aims. Some come as a surprise to the reader.

How was the qualitative data analysis made? How many responders adhered to each of the presented statements?

What were the sizes of group differences?

Were there significant group differences for informal caregivers?

Discussion

The effect sizes are needed to see to what extent the treatment was effective. Note that the comparison group was almost a waiting list, so no improvement would be expected. However, there were no group differences in a number of outcomes (pain interference, depression, anxiety, and stress) and the size in those that were different is unknown.

Can the authors try to explain why changes were not observed in a number of outcomes and how this should make the treatment change?

Author Response

Please see the attachment.Thanks for your comments and suggestions.

Reviewer 2 Report

1. Manuscript represents a randomized controlled trial of DPM for older adults with chronic pain.

2. Abstract: add p values for significance associations stated.

3. The sentence beginning on line 77 replaced in Methods: intervention.

4. Provide the actual p values in the body of text.

5. For clarity, rephrase the sentences beginning on lines: 95, 325, 330, 332.

6. Figure 1: relabel as Flow Diagram

                notation should be: n=32 (100%), n=3 (9%), etc.

7. Left – justify the columns in figures 2, 3 and tables 2-7.

8. Table 7: add a column of p values

                check typo, mean changes column, lines 2, 4

9. Conclusion: 

Place weaknesses in discussion above "strengths"

Amplify significance of findings and clinical implications as a model of care and caregiver support

Amplify recommendations for future research

Author Response

Please see the attachment. Thanks for your comments and suggestions.

Reviewer 3 Report

Thank you for giving me the opportunity to review this interesting RCT. It is well written, timely and concise. However, I have some suggestions for the authors:

Major concerns:

  • Was the normality of the data checked by Shapiro-Wilk or Kolmogorov-Smirnov tests?
  • Specified the Intention-to-treat method used in this article. 
  • It seems that paired t-tests were used to examine group*time interaction. If it is right, it is not the correct approach. Authors should have used a repeated measured ANOVA, reporting the group*time interaction (since they have 2 groups and they have two measurements - T0 and T1-). 
    • "To explore intervention efficacy, which compared the changes in the mean scores of the 219 outcome variables within-group at pre- and post-intervention, paired t test was used".
  • This consideration of the ANOVA should be done to both participants and caregivers.
  • Limitations should be recognized, not only strengths.

Minor issues:

  • Improve the quality of Figure 1. 
  • Could you better explain the rationale behind the correlations between digital use frequency and the rest of the outcomes? Readers would appreciate it. 
  • I encourage the authors to place the tables closer to the first time they have been cited rather than at the end of the results section.

Author Response

(The authors gave the same response as above.)

Round 2

Reviewer 1 Report

The authors have responded to some of the concerns raised. However, there are still important flaws in the manuscript. See my comments below:

Abstract

participated in “an” 8-week

Totally à In total,

The numbers provided after “pain score was significantly reduced”, “self-efficacy compared with the control group, “quality of life… significantly improved”, and “increased exercise time” should be the estimates, CIs, p values, and effect sizes

“And the elderly significantly and reported developing good exercise habits.” Only the

The conclusions should provide insights about the study, not about the need to replicate the findings.

Introduction

The first sentence is too long. Please make English be revised by a native speaker

“Pain prevalence among community-dwelling older adults.” Is this acute or chronic pain? How come the differences in prevalence are so large, from 25% to 90%?

“the main objective is to relieve the pain and pain-related symptoms, rather than to eliminate the pain thoroughly.” The sentence is not very compelling. I understand that the authors mean that, because eliminating pain completely in this population with comorbidities might be challenging, reducing it to some extent and also very importantly improving other outcomes might be more feasible. This should be more clearly specified for the non-expert reader and examples of recommended pain-related outcomes should be provided (i.e., mood, quality of life, functioning, etc.).

The sentence “The pain relief for the 56 elderly can be less than expected [19]” should be better linked with the previous sentences.

reported that the elderly “who” participated

Define “CDC” before its first use.

face-to-face programs

The sentence “Pain education made older adults more clearly with their responses to pain” is very confusing.

Where changes in pain explained by changes in self-efficacy (i.e., mediation)?

There is something missing in the sentence “aging people's ability of daily”

Is the sentence “Poor attendance rate and poor compliance with the intervention are to the disadvantage of pain management” related to older patients?

How is exactly the program dealing with reported barriers, such as low literacy, poor memory, difficulties in understanding theoretical knowledge, difficulties in developing good exercise, cognitive impairment, and low acceptance of technology?

In the paragraph starting with “To improve the above problems”, several sentences appear to be poorly interconnected.

“Explore the perspectives” is a very unspecific goal.

Data analysis

“using a conventional content analysis” is too vague. How many researchers conducted the analysis and what was their level of agreement on the number and label of categories?

Results

The authors state that “No significant change in depression subscales, anxiety subscales and stress subscales was found between the two groups at two time-point comparisons”

What are the results being interpreted in the results section, the between-groups or the interaction effects? The values reported in the text do not match

Discussion

The authors state that “The result of the significant decrease of pain intensity and pain interference are in keeping with previous studies.” However, there were no treatment effect on interference in the present study.

The authors state that “The teaching materials used in program are focusing on theoretical level, which are difficult to understand, even for some informal caregivers.” This means that the program is not good and should be changed?

The authors state that “Most informal caregivers provided feedback that they were also suffering from chronic pain and got benefit from the pain management.” In what sense? Only informal reports? Why mentioning caregiver burden if this was not evaluated?

Most of the conclusions are focused on secondary study results (technology) or on aspected not evaluated in the study (impact on caregiver). There is little discussion about why the treatment worked for some outcomes and not for others.

Tables

Table 2 is very confusing. I suggest changing the positions of time and group condition.

When p=0.000, please state p<.001. No need to include “0” before the decimal in p values.

In Table 3, change “within times” to “within groups”

Table 3 is poorly organized (see QoL results specially) and results on average exercise are missing.

English requires revision throughout the tex

Author Response

Thanks for all your comments. Please see the attachment.

Reviewer 3 Report

I encourage authors to:

  1. Revise the format of tables and place them close to the text where they were cited.
  2. The intent-to-treat approach still remains unclear. 
  3. Correct the p-values. P-values written like "0.000" should be replaced for <0.001.

Author Response

Thanks for all your comments and suggestions. Please see the attachment.
